# Soil and Vegetation Development on Coal-Waste Dump in Southern Poland

**DOI:** 10.3390/ijerph19159167

**Published:** 2022-07-27

**Authors:** Oimahmad Rahmonov, Agnieszka Czajka, Ádám Nádudvari, Maria Fajer, Tomasz Spórna, Bartłomiej Szypuła

**Affiliations:** 1Institute of Earth Sciences, Faculty of Natural Sciences, University of Silesia, 41-200 Sosnowiec, Poland; oimahmad.rahmonov@us.edu.pl (O.R.); adam.nadudvari@us.edu.pl (Á.N.); maria.fajer@us.edu.pl (M.F.); bartlomiej.szypula@us.edu.pl (B.S.); 2Institute of Social and Economic Geography and Spatial Management, Faculty of Natural Sciences, University of Silesia, 41-200 Sosnowiec, Poland; tomasz.sporna@us.edu.pl

**Keywords:** soil–vegetation link, soil features, vegetation succession, coal dump, post-industrial sites

## Abstract

As an anthropogenic element of urban landscapes, coal heaps undergo changes due to both natural and anthropogenic factors. The aim of this study was to determine the common development of soil under the influence of vegetation succession against a background of environmental conditions. Vegetation changes and soil properties were analysed along a transect passing through a heap representing a particular succession stage. It was found that changes in the development of vegetation were closely related to the stages of coal-waste disposal, where the initial, transitional, and terminal stages were distinguished. The mean range of pH (H_2_O) values in the profiles was 6.75 ± 0.21 (profile 1), 7.2 ± 0.31 (profile 2), 6.3 ± 1.22 (profile 3), and 5.38 ± 0.42 (profile 4). The organic carbon (OC) content in all samples was high, ranging from 9.6% to 41.6%. The highest content of total nitrogen (Nt) was found (1.132%) in the algal crust and sub-horizon of the organic horizon (Olfh-0.751%) and humus (A-0.884) horizon in profile 3 under the initial forest. Notable contents of available elements were found in the algal shell for P (1588 mg∙kg^−1^) and Mg (670 mg∙kg^−1^). Soil organic matter content was mainly dominated by *n*-alkanes (*n*-C_11_-*n*-C_34_) and alkanoic acids (C_5_–C_20_). Phytene and Phytadiene were typical for the algal crust on the initial pedigree. The initiation of succession was determined by the variation in grain size of the waste dumped on the heap and the variation in relief and associated habitat mosaic. Algal crusts forming on clay–dust mineral and organic material accumulating in the depressions of the site and at the foot of the heap can be regarded as the focus of pedogenesis.

## 1. Introduction

The extraction of mineral resources contributes to economic development, yet their exploitation also leads to serious and often irreversible changes in natural systems [1,2,3]. These changes are most visible in ecological systems in the form of the complete degradation of soil and plant cover [4,5]. Hard-coal mining and related technological processes bring about long-term changes in the environment, as well as in the landscape of mining areas and those in their vicinity [6,7,8,9,10,11].

Coal-waste dumps are subject to processes of overgrowth over time, both through natural succession and planned reclamation. The spontaneous succession of vegetation on waste heaps resulting from coal exploitation has long aroused the interest of naturalists around the world [7,12,13,14,15,16,17,18,19,20,21,22], because these are places where plant-succession processes [9,18,23] and soil development [5,24,25,26,27,28] can be observed and monitored from the moment of initiation. The course of succession stages of vegetation on different types of heaps occurs slightly differently in time and space [15,22], which stems from the method of storing burnt material and the size, shape, and age of the heap [6,9,16,20,29,30]. Vegetation diversity and changes in floristic composition as an effect of succession are examined on the basis of the type of material stored, water relations in the heap surroundings, and the diversity of granulometric composition [15,22,30], which is important in the early stages of succession and soil development in extreme environments [27]. The formation of vegetation cover as a final effect of succession prevents many negative environmental effects (slope erosion, dust emission, and migration of heavy metals and organic pollutants), hence spontaneous succession is often treated as one of the ways to revitalize post-mining lands [22,31].

With the initiation of plant succession, soil-formation processes begin. Unburnt coal shale provides good soil-forming material. Soil formation on mining waste heaps is much faster compared to natural soils. In the initial stages of formation, the main pedogenetic process is intensive humification [24], which is dependent on the diversity and source of organic matter [27,32,33]. Poorly developed topsoil can be formed within a dozen or so years of pedogenesis, and on reclaimed heaps even within the space of a few years [34]. Owing their presence on the surface to human activities, natural fragments of bedrock are considered industrial artefacts [35]. These rocks are the parent material for the emerging soil, and their nature determines the rate and characteristics of the forming soil concurrently with the developing initial ecological system.

In soils in the early stages of development, the plant food comes from components released by the weathering of the bedrock and from components formed from the decaying of plants that grow on such surfaces. After these have decomposed, nutrients essential for plants are released. This influences the rate at which the area is colonised, especially by plant species with a wide ecological tolerance to the area in question, such as *Calamagrostis epigejos* and *Chamaenerion palustre*. By determining their chemical composition in terms of the content of the main elements (Ca, K, Na, Mg, P, Fe, and Al), we can recognise their potential to enrich the initial soils with nutrients, while the analysis of heavy metals determines their accumulation potential.

The objectives of this study were: (i) to determine the common development of vegetation succession, (ii) to establish its influence on soil-formation processes and soil features in relation to the environmental conditions, and (iii) to determine the chemical composition of plants as an element enriching soils with nutrients in the early stages of succession on a post-mining coal-waste heap.

## 2. Materials and Methods

### 2.1. Study Site Description

The study was conducted at the mining-waste dump of the Szczygłowice Coal Mine (50°11′30″ N, 18°38′35″ E) in Knurów in the NW part of the Upper Silesian Coal Basin in southern Poland (Figure 1). The deterioration dump was created in an area degraded by mining exploitation. As a result of subsidence, the area has sunk by about 20 m [36].

In the oldest part of the dump, stretching along the riverbed of the River Bierawka, waste material was first stored in 1977. In the following years, the heap expanded in an easterly direction. Part of the reclaimed area was covered with deposited waste. In 2002–2003, the SW part of the heap was afforested (Figure 1A). The height of the heap is about 12 m above sea level and its area about 45 ha (45,000 m^2^), of which the reclaimed area is 0.75 ha [36].

Before the drastic environmental degradation, potential natural vegetation compatible with geological and soil conditions developed in the area in the form of deciduous forests of the varieties *Tilio-Carpinetum*, *Dentario-ennephyllidis*, *Fraxino-Alnetum*, and *Leucobryo*-Pinetum on the sandy habitats [37]. Today nothing remains of these potential communities in the neighbourhood. Part of the area around the study site was recultivated towards forestry through the introduction of *Pinus sylvestris*, *Betula pendula*, and *Robinia pseudacacia*.

### 2.2. Plant Sampling

Vegetation studies concerning the floristic investigation were carried out in 2020 in the beginning of the growing season (May/June) and the second full vegetation period (July/August). The floristic examinations consisted of an inventory of plant species growing on the analysed area in divided surfaces (Figure 1A). Raunkiaer’s plant life forms were defined (Appendix A).

The course of vegetation succession in the area of the coal-waste heap was examined along a transect (Figure 1A (1–4) and (I–IV)). It ran from the newest areas of heaps with freshly dumped post-mining waste, through sodding areas, to the emerging initial forest with a dominant share of *R. pseudacacia*. Vegetation changes were analysed on both the basis of satellite imagery (Landsat 4-5TM, Landsat 7ETM+, Landsat 8 OLI) and direct field survey, including an analysis of drone images. The Normalized Difference Vegetation Index (NDVI) was calculated from Landsat images from 1985 to 2020. In the areas of succession stages that were identified, a floristic list (Appendix A) was compiled, and plant and soil samples were collected for laboratory analysis.

### 2.3. Plant Chemical Composition

In order to understand the potential enrichment of the substrate in nutrients, the chemical composition of selected dominant plant species (*R. pseudacacia*, *Calamagrostis epigejos*, *Chamaenerion palustre*) was studied, assuming that after death they would provide the initial soil with mineral nutrients. As plants capture or retain wet and dry precipitation on their organs, this was also treated as a source of components, and therefore the test parts were not washed in distilled water before analysis. In the chemical composition of the plants, the same elements were studied as in the soils.

Leaves, stems, seeds, and roots of *R. pseudacacia*, *C. epigejos*, and *Solidago canadensis* were sampled at the end of the 2020 vegetation season in September and early October. The above-ground parts of *C. epigejos* and *Ch. palustre* were examined as a whole, and were not divided into stem and leaves.

### 2.4. Soil Sampling

Three soil profiles were made under dominant plant communities representing particular stages of succession: initial profile 1, optimal profile 2, and terminal profile 3, (Figure 1A (1–4) and (I–IV)). In the initial stage of succession under algocenoses (mainly with contribution from blue-green algae), 5 soil samples were also taken. An organic mineral crust is formed under these communities (Figure 1A (4)). A total of 14 soil samples were taken for laboratory analysis. Samples for testing were collected during June 2020–April 2021.

In the laboratory, air-dried soil samples were sieved (<2 mm) and analysed, following the standard procedures of Bednarek et al. [38], namely, pH measured potentiometrically in H_2_O and in 1N KCl using a glass electrode, total organic C (%) according to Tiurin’s method, total nitrogen (Nt) content using the Kjeldahl method, and hydrolithic acidity (Hh) according to the Kappen method. Total phosphorus (Pt) was measured using the Bleck method, modified by Gebhardt, available phosphorus (Pavail) by Egner–Riehm’s method, and available Ka-avail and Mgavail by PN-R-04023/23 [38].

The preliminary preparation of the samples for analysis involved drying at room temperature and 105 °C, and homogenization. The total composition of main elements—Ca, Mg, K, Na, P Fe, S, Al, Zn, Hg, As, Cr, Cu, Ni, Co, Cd, and Pb—and in plant material and soil were measured using ICP-OES (inductively coupled plasma optical emission spectrometry) after wet mineralization in nitrohydrochloric acid (3HCl + HNO_3_). The analyses were performed in the ACME Laboratory (Vancouver, Canada) using AQ250_EXT (soils) and VG105_EXT (plant tissues) procedures and 5 g samples. All plant tissue and soil samples were analysed in triplicate for all the parameters being investigated, and mean values were calculated.

The granulometric composition of the samples was determined using standard grain size analysis with a fixed mesh size sieve column. The test was carried out with a set of sieves with different mesh sizes: 20 mm, 10 mm, 5 mm, 2 mm, 1 mm, 0.5 mm, 0.25 mm, 0.1 mm, 0.05 mm, 0.02 mm, 0.006 mm, and 0.002 mm. The mass of the sample remaining in each sieve was calculated as the percentage of grains of a given size in the total mass of the sample.

### 2.5. Statistical Analyses

Spearman’s rank-correlation coefficient was applied to check whether there was any relationship between the concentrations of the selected elements in plant materials, soil materials, and crust with algae in the study profiles. The exact values of the correlation coefficient were calculated for alpha = 0.05. Significant differences in the measured soil chemical parameters among the study profiles were estimated using the Kruskal–Wallis test. All statistical analyses were performed using Statistica 14.0.0.15 software.

### 2.6. Soil Organic Matter

The powdered samples (c. 18–20 g) were extracted using dichloromethane (DCM) and methanol (1:1) with a Dionex ASE 350 solvent extractor. The extracted organic matter was then analysed by gas chromatography–mass spectrometry (GC-MS) using an Agilent Technologies 7890A gas chromatograph and Agilent 5975C network mass spectrometer with a triple-axis detector (MSD) at the Institute of Earth Sciences in the Faculty of Natural Sciences at the University of Silesia. The results were obtained in full-scan mode and processed with Hewlett-Packard ChemStation software. The compounds were identified according to their mass spectra (using peak areas acquired in the manual integration mode) by comparing peak retention times with those of standard compounds, interpreting MS fragmentation patterns, and mass spectral databases using GEOPETRO NIST17 (National Institute of Standards and Technology, Gaithersburg, MD, USA) and Wiley (W10N11) [39,40].

## 3. Results

### 3.1. Vegetation Changes

Based on satellite images (Landsat 4-5TM, Landsat 7ETM+, and Landsat 8 OLI) using the NDVI index, changes in heap shape and vegetation extent were identified (Figure 2). The highest value of NDVI was in the last decade and stood at 0.8, indicating a higher degree of vegetation colonisation (compare Figure 1).

Two types of succession were distinguished: primary and secondary. Within the areas with primary succession, three stages were distinguished: initial, optimal, and terminal (Figure 1A (I–IV) and Figure 3). The direction of succession was determined by the degree of spatial differentiation of the grain size of the material stored in the heap. The spatial distribution and distribution of the stored waste was also influenced by morphological processes (surface runoff, aeolian erosion).

### 3.2. Primary Succession

In the initial stage, algae and cyanobacteria (algal crusts) initiated succession in the areas of the heap where organic mineral material had accumulated in depressions or at the foot of convex forms (Figure 1A (4)). This provides favourable conditions for the encroachment of species under conditions of moisture abundance. They were visible on the surface in the form of a green overlay (Figure 1A (4)). When moisture is lost, a hard crust forms from the dried fine sediments. Some of these crusts crack after drying. Algae communities colonised ground surfaces dominated (%) by very fine sand (0.05 < d ≤ 0.2 mm), dusts (0.002 < d ≤ 0.05 mm), and clay fractions (d ≤ 0.002 mm).

On younger surfaces, post-coal waste does not undergo weathering processes as quickly (Figure 1A (I–II)). These surfaces are characterised by heterogeneous morphology, dominated by a coarse-grained, stony fraction, and spatial distribution often results from the way the material was stored on the heap. Such a substratum is first colonized by the pioneer species Ch. palustre, individually accompanied by *Oenothera biennis*, *Erigeron canadensis*, and *Daucus carota*.

On plots with a significant share of fine-earth fraction (0.1 < d ≤ 0.2 mm) succession was initiated mainly by *C. epigeojos* and *S. canadensis*, which often formed single-species plant communities (Figure 1A (III)). Between the clumps, *C. epigejos*, *P*. *syl**vestris*, and *B. pendula* occurred singly, which indicated the direction of forest formation. From herbaceous plants at this stage, *D. carota*, *Corynephorus canescens*, *Ch. palustre*, *Echium vulgare*, and others were found (Appendix A). The species diversity is due to the variation in macro- and micro-relief and the associated habitat mosaic. This is a transitional phase of succession, which can persist for a long period, and eventually transforms into mixed forest.

### 3.3. Secondary Succession

A separate direction of succession was the development of a black locust forest, which was artificially introduced on a prepared substrate with a 23 cm-thick overlay of soil characterised by good water and air conditions (Figure 1A (IV)). Here, a *R. pseudacacia* forest with *P. sylvestris*, *B. pendula*, and *Populus nigra* had developed. The ground cover of this forest was composed of species with a wide range of ecological tolerance, including *S. canadensis*, *S. virgaurea*, *Tanacetum vulgare*, *Trifolium pratense*, *T. arvense*, *Erigeron annuus*, *Achillea millefolium*, *E. vulgare*, *Hieracium pilosella* (compare Appendix A), and *Bryum caespeticum*.

### 3.4. Flora Diversity

A total of 38 species of vascular plants belonging to 38 families and 33 genera were found in the area studied. The number of species varied within the investigated sites (Appendix A), resulting from the age of the sites where post-mining waste had been dumped. The oldest site (3) had a higher species diversity (30 species), while Site 2 had 21 species and Site 1 was represented by 4 species. Within families, most species were represented by *Compositae* (13 species), *Poaceae* (5), and *Leguminosae* (4), respectively. In terms of life forms, hemicryptophyte dominated in all the plots. A small proportion was megaphanerophyte (9.52%-Site 2 and 16.6-Site 3) and therophyte (4.7%-Site 2 and 6.6%-Site 3). 

### 3.5. Soil Morphology

The analysed soil was defined as the technosol type, urbic technosol subtype, and hyperskeletic spolic technosol subtype, according to the IUSS Working Group WRB [41]. These types are determined on the basis of their artificial origin and contain a large number of artefacts. The waste on which the soil-forming process takes place is typical for the Upper Silesian Coal Basin and consists mainly of clays, with lesser amounts of carboniferous siltstones and sandstones, which influence the soil chemistry.

In the initial stages of succession (profiles: 1 and 4), the formation of soil horizons is not observed (Table 1) and the substrate has a stratified character, resulting from the overlaying of the waste stages (C1 and C2 denote successive layers). In the case of profile 4, the presence of an organic mineral horizon (OC) associated with the colonization of blue-green algae was observed (Figure 1A (4)). The formation of organic (O) and humus horizons (A) was found in a grouping with C. epigejos and initial forest with *R. pseudacacia* (Table 1). Contemporary vegetation plays an essential role in the formation of the organic humus horizon in contrast to weathering processes.

### 3.6. Soil Physico-Chemical Features

Particle-size analysis revealed the differentiation of the particular samples in terms of grain size (Table 1). The biggest proportion (from 58% to 85%) of the gravel fraction (20 < d ≤ 76 mm) and very coarse sand (1.0 < d ≤ 2.0 mm) was found in profiles 1, 2, and 3 (except for Olf level). A completely different fractional composition of samples derived from organic soil crust formed with the participation of blue-green algae (profile 4). The predominant fraction was the dusty fraction (0.002 < d ≤ 0.05 mm), and its share here was from 48% to 79%. The clay fraction was also significant (<0.002 mm): 7–25%.

The soil reaction varied from strongly acid: 4.8 pH (H_2_O) and 4.1 pH (KCl) in the humus level in profile 3, and 4.9 pH (H_2_O) and 4.8 pH (KCl) in profile 2 through neutral (profiles 1, 4) to basic (Table 2). In turn, hydrolytic acidity (Hh) ranged from 1.60 to 5.36 cmol(+)/kg, with the highest values found mainly in the plant root zone. The average pH (H_2_O) values in profiles were: profile 1 = 6.75 ± 0.21, 2 = 7.2 ± 0.31, 3 = 6.3 ± 1.22, and 4 = 5.38 ± 0.42. On the other hand, pH KCl, the reaction averaged 6.6 (±0.14, profile 1), 6.9 (±0.3, profile 2), 6.0 (±1.08, profile 3) and 4.9 (±0.56 in profile 4). Significant differences on the Kruskal–Wallis test were observed between profiles 2 to 4 for the reaction performed in water (*p* = 0.0429) and in KCl (*p* = 0.0359).

Analysis of roasting losses indicated a high content of organic matter in the substrate, which is mainly derived from landfill coal refuse. The organic carbon (OC) content in all samples was high, ranging from 9.6% (profile 4/2) to 41.6% (profile 2/ACq). The highest content of total nitrogen (Nt) was found (1.132%) in the algal crust, organic with suborganic (Olfh-0.751%), and humus (A-0.884) horizon in profile 3 under the initial so called *Robinia* forest (Table 2).

The C/N ratio was highly variable and ranged from 8 to 99. This range indicates very weak or even non-existent microbial activity. The case of the narrow ratio in profile 4 in sample 2 (OC, which is 8) was related to dissolved and transported organic matter, while in profile 3 in the humus horizon (C/N-12), it was related to the transformation of overburden humus under the *R. pseudacacia* canopy. The prevailing xeric and thermal conditions on the heap promoted the decomposition of organic matter.

The results obtained from the analysis of bioavailable Mg, K, and P contents in the profiles studied are presented in Table 2. The average content of magnesium was 397 mg·kg^−1^ (263.5–670 mg·kg^−1^), potassium 187 mg·kg^−1^ (132–277 mg·kg^−1^), and phosphorus 3.82 mg·kg^−1^ (0.60–16.20 mg·kg^−1^). The highest contents of these bioavailable elements were found in the organic mineral crust of algocenoses (Table 2, profile 4).

The content of total phosphorus (Pt) as an indicator of anthropogenic soil pollution ranged from 455 to 1588 mg·kg^−1^ in the algal shell, with a mean value of 1177 mg·kg^−1^. In profiles 2 and 3, significant statistical differences were found in the total Pt (Pt) content at the *p* = 0.0283 level (Kruskal–Wallis, at α = 0.05).

Based on Spearman’s rank correlation coefficient, within the study plots (salt profiles) we found very strong positive correlations of KCl-K avail. (r(12) = 0.7181, *p* = 0.002), C/N ratio-Corg. (r(12) = 0.8571, *p* ≤ 0.000), P avail., and total phosphorus (Pt) (r(12) = 0.6028, *p* = 0.017), among others. In contrast, a very strong negative correlation was found between soil pH (H_2_O) and hydric acidity (Hh: r(12) = −0.6149, *p* = 0.014), among others. A similar situation was found between C/N and total nitrogen (Nt: r(12) = −0.7464, *p* = 0.001). These correlations between particular soil physico-chemical properties are often due to the character of the stored mine tailings (non-selective storage) and not to the processes of plant–soil cover development (Appendix A). 

### 3.7. Soil Organic Matter Composition

*n*-Alkanes (*n*-C_11_-*n*-C_34_) and *n*-alkanoic acids (C_5_-C_20_) were predominant, although their distribution varied according to the developed vegetation cover (Figure 2) (Appendix A). Typically, *n*-alkane predominance over alkanoic acids was observed in those samples where the soil cover was less developed, e.g., profile 3-ACq-horizon and profile -C1-horizon (32.6–83.3) (Appendix A). *n*-Alkanoic acids were noted (ranging from C_5_ to C_20_) with elevated relative percentages only in the A horizon (profile 3) and the soil algal crusts (Figure 1A (4), profile 4). The CPI indices (Appendix A) also clearly indicated the soil-developing processes, i.e., CPI(*n*-C_24_-*n*-C_34_) ~1.0–1.1 characterize typical matured fossil organic material from the Upper Silesian Coal Basin, e.g., Nádudvari et al. [42] or Nádudvari and Fabiańska [43], less influenced by recent organic material in profile 2 (A) and profile 1 (A) (Appendix A).

The biodegradation and water-washing of short-chain *n*-alkanes were also (1.15–1.98) compared to profile 1 (C_1_-C_2_-horizon) in the soil samples using short-chain/long-chain abundance (Appendix A). In the topsoil (A), this ratio was lower (affected by leaching and biodegradation) than in the B and C layers. The algal crust mostly showed bimodal distribution of *n*-alkanes with maximal peaks in short-chain *n*-C_17_ and *n*-C_18_ and in long-chain *n*-C_27_–*n*-C_31_ peaks.

In samples (profile 3: A-Cq horizon; profile 2: A-horizon) and in the algal crusts, δ- and γ-Tocopherols were identified. Elevated percentages were only in the algal crusts, especially γ-Tocopherol. With regard to the presence of sterols, there were large differences between the sample types, i.e., Cholestanol, 5β-Cholestan-3β-ol, Cholest-5-en-3β-ol (Cholesterol) and steroids Cholestan-3-one and Cholest-4-en-3-one were only identified in algal crusts with a predomination of 5β-Cholestan-3β-ol and occurred in higher concentrations than other sterols (Appendix A). Among the phytosterol biomarkers Stigmastanol, Stigmast-5-en-3β-ol (Sitosterol), 5α-Stigmastan-3-one, Stigmasta-3,5-dien-7-one, and Stigmast-4-en-3-one (Sitostenone) were also present in samples (Appendix A) where soil forming occurred, e.g., 3 profile in A horizon in addition to algal crusts. Styrene, Friedelan-3-one, and Benzoic acid (lignin degradation product) were commonly present in Profile 3–Cq and in algal crusts, Methoxyphenol and Methyl-oleanonate were present only in the A horizon of profile 3. The series of β-Amyrone, β-Amyrine, α-Amyrine, and α-Amyrin acetate are typical anigosperm biomarkers, although high percentages were identified only in those samples where soil cover was present or in algae crusts. These compounds are not algae-related, as they come from other sources [44,45]. Among the organic sulphur compounds (OSC), 3β-Cholestane-3-thiol, Benzenemethanethiol (Benzyl mercaptan), 4-Methylbenzenemethanethiol (4-Methylbenzyl mercaptan), 3-methyl-2-(3,7,11-trimethyldodecyl)thiophene, *3-n*-Hexadecylthiophene were present only in algal crusts (Figure 4). Typically, plant- (and soil)-related compound Methyl benzoate was commonly identified in many soil samples (it is far less prevalent in algal crusts), and indicates bioactivity in the dump soil.

Vanillin is also an important biomarker for the identification of soil formation and bioactivity; therefore, this compound, together with other Hydroxy-methyl-benzaldehydes, was identified in most of the samples. The following compounds, alkenes, Phytene and Phytadiene isomers, Neophytadiene, alkenes, and Phytol (Figure 4) were present only in algal crusts as a result of biochemical reactions, and even in algal crusts they were identified despite these samples being in an initial state of crust forming from the dump. Among the fatty alcohols, Tetradecan-1-ol and 10-Dodecen-1-ol were present in algal crusts. The *n*-C_17_–*n*-C_24_ alkenes occurred in higher percentages in algal crusts, especially *n*-C_17_ alkene. Other coal-waste-related pollutants were identified, e.g., Benzaldehyde, Acetophenone, and Indene (Appendix A).

### 3.8. The Major Element Composition of Ch. palustre, C. epigejos, R. pseudacacia, and Soil

Table 3 presents the results of an analysis of the major elements in the plant and soil materials studied. The above-ground parts (stem, leaves as a mixture/mix) of *Ch. palustre* were dominated by Ca (7000 mg·kg^−1^), K (6200 mg·kg^−1^), and Mg (3480 mg·kg^−1^), and Al was found in the lowest concentration (Table 3). Higher concentrations of the above elements (except sulphur) were found in the roots of *Ch. palustre*.

In the chemical composition of the soil material (profile 1) under *Ch. palustre*, the contents of Ca, K, Mg (C1 -0-14 cm), and P (C1 -0-14 cm) were lower than in above-ground parts (Table 3), while the opposite was true for iron, sulphur, and aluminium. The chemical composition of the soil at different depths varied and was due to the way the material was stored and its granulometric composition, where silty clay fractions make up a significant proportion (Table 1). 

In the case of *C. epigejos*, its above-ground tissue was dominated by K (9700 mg·kg^−1^), while the Ca content was 1400 mg·kg^−1^ and Mg 850 mg·kg^−1^. The Ca content in the roots of *C. epigejos* was the lowest (1100 mg·kg^−1^), in contrast to *Ch. palustre* (8400 mg·kg^−1^) and *R. pseudacacia* (3800 mg·kg^−1^). On the other hand, the highest concentrations of Fe (12,610 mg·kg^−1^), phosphorus (8600 mg·kg^−1^) and aluminium (4000 mg·kg^−1^) were determined in the roots of this species. In both *Ch. palustre* and *C. epigejos*, higher elemental concentrations were found in the roots. This is connected with the fact that they are perennial species and the roots are alive, while the above-ground parts of the analysed species die every year, hence the differences indicated.

The chemical composition of the soil material varied under *C. epigejos* (Table 3). On average, in all the levels analysed, the content of the elements was as follows: Ca = 3800 mg∙kg^−1^, K = 1600 mg∙kg^−1^, Mg = 2966 mg∙kg^−1^, P = 316 mg∙kg^−1^, Fe = 17,100 mg∙kg^−1^, S = 6633 mg∙kg^−1^, and Al = 5533 mg∙kg^−1^. Similarly, the contents of Fe and Al in the profiles indicated a low degree of weathering of the heap material, their homogeneity, and that they were components of post-mining waste. On the other hand, relatively higher contents of Ca and Mg may stem from decomposition of both above-ground and underground parts of plants.

The highest amounts of Ca were found in *R. pseudacacia* in bark (16,800 mg∙kg^−1^), leaves (12,400 mg∙kg^−1^), roots (3800 mg∙kg^−1^), and pods (2900 mg∙kg^−1^). The leaves also showed the highest contents of K (13,700 mg∙kg^−1^) and Mg (3140 mg∙kg^−1^). In pods, K was also dominant and its content was 9300 mg∙kg^−1^. The highest content of potassium was found in the leaves (13,700 mg∙kg^−1^) and roots (11,600 mg∙kg^−1^). In all the plant fractions investigated, the lowest content was noted for Na (Table 3).

The composition of the total soil material did not differ greatly from the previous profiles, except for the calcium content, which was generally lower, in the range of 1000 to 1600 mg∙kg^−1^. A similar pattern applied to all elements (Table 3), because the parent rock for the developing soils in the coal heap areas is post-coal waste with a similar chemical composition.

Based on Spearman’s rank correlation coefficient, a very strong positive correlation between Fe-Al (r(6) = 0.8728, *p* = 0.0004) and Fe-Na (r(6) = 0.8263, *p* = 0.0114) was found in the chemical composition of the plants studied in terms of major elements. Other elemental relationships are presented in Appendix A.

Compared to the plant materials, positive correlations were found within the soil material between Ca and Mg (r(6) = 0.7904, *p* ≤ 0.019), Ca and Na (r(6) = 0.7281, *p* = 0.040), and Na and S (r(6) = 0.8476, *p* = 0.007), and a negative correlation between S and Al (r(6) = −0.8571, *p* = 0.006) (cf. Appendix A).

### 3.9. The Heavy Metal Content of Ch. palustre, C. epigejos, R. pseudacacia, and Soil

The results of the analysis of heavy metal content in the plant and soil material are presented in Table 4. The content of Zn in all the plant species studied showed variations, both in roots and in above-ground plant parts. The highest concentrations were found in the roots of *C. epigejos*, where it was 70 mg∙kg^−1^, and in the case of roots of *R. pseudacacia* and *Ch. palustre* almost identical values were obtained: 35.3 and 35.4 mg∙kg^−1^. In the green parts of plants, its range was from 14.8 to 65.4 mg∙kg^−1^ (Table 4).

The Pb content in the above-ground parts of the plant species in question ranged from 0.98 mg∙kg^−1^ (*C. epigejos*) to 2.89 mg∙kg^−1^ (*Ch. palustre*). The highest contents of Pb were recorded in the roots of *C. epigejos* (21.16 mg∙kg^−1^) and *R. pseudacacia* (5.81 mg∙kg^−1^). Similar patterns were observed for Cu content. The lowest contents within the analysed elements were noted for Cd (0.08–1.12 mg∙kg^−1^) and Hg (Table 4).

The contents of heavy metals in all analysed sites (soil profiles) did not show significant differentiation and had similar values, which indicates the similarity of the chemical composition of the stored coal waste. Their content was basically not reflected in the chemical composition of plants. Slightly higher values were marked in humus horizons for Zn, Pb, Ni and Cu. 

The contents of major elements in the algal thallus (organic mineral, OC) were higher than in the materials from the soil profiles (Table 5). The mean macronutrient contents of the thallus studied were: 7560 mg∙kg^−1^ Ca, 1860 mg∙kg^−1^ K, 372 mg∙kg^−1^ Na, 4140 mg∙kg^−1^ Mg, 1834 mg∙kg^−1^ P, 24,340 mg∙kg^−1^ Fe, 2880 mg∙kg^−1^ S, and 10,780 mg∙kg^−1^ Al. The phosphorus content was notably higher than in the coal waste itself, and this is related to its organic origin.

A completely different content of heavy metals was found in the case of algal crusts (often called biological soil crusts), consisting mainly of a silty clay fraction. As already mentioned, this crust is formed by the transport of mineral and organic substances during heavy rainfall. When the carrying force of the water decreases, the transported material is deposited. As this sediment consists mainly of fine and clay fractions, it promotes water retention. Over time, as the water evaporates, this material turns into a hard crust (algae are the first to colonise and hence the name) that is not subject to water erosion. Such fragments can be considered the focus of pedogenesis in areas of coal heaps. 

Within the 5 algal crusts analysed, mean values of 457.6 mg∙kg^−1^ Zn, 54.7 mg∙kg^−1^ Cu, 52.8 mg∙kg^−1^ Pb, 33.2 mg∙kg^−1^ Ni, 33.6 mg∙kg^−1^ Cr, 15.5 mg∙kg^−1^ Co, 8.8 mg∙kg^−1^ As, 2.5 mg∙kg^−1^ Cd, and 0.158 mg∙kg^−1^ Hg were determined (Table 6). The highest contents of heavy metals within the crusts were found in sample 4. This was a multilayered crust, indicating multiple depositions of material.

On the basis of Spearman’s rank correlation coefficient, very strong statistically significant correlations were found between Cu and Cr (r(6) = 1.9461, *p* ≤ 0.000), Cu and Zn (r(6) = 0.7857, *p* = 0.020), Co and Ni (r(6) = 0.9285, *p* ≤ 0.000), and Pb and Cd (r(6) = 0.8809, *p* ≤ 0.003) in the chemical composition of the plants with regard to their heavy metal contents. The exception was Sr (no statistically significant correlations). Compare Appendix A.

In the soil material, a positive strong correlation was found between the following elements: Zn and Cd, Pb and Cu, Pb and Sr, Cu and Sr, As and Hg, As and Ni, and Cr and Cd. Negative correlations were found between Cu and Cr (r(6) = −0.9819, *p* ≤ 0.000), Pb and Cr (r(6) = −0.8365, *p* = 0.009), and Cd and Cu (r(6) = −0.7324, *p* = 0.0387). Other elemental relationships are shown in the Appendix A.

## 4. Discussion

### 4.1. Habitat Conditions and the Course of Succession

The process of vegetation colonisation is closely connected with the stages of heap compaction. In the area of the analysed heap, three stages of succession were distinguished along the determined transect: initial, transitional, and terminal (Figure 1 and Figure 2). The initiation of succession in this stage is conditioned by the presence of depressions and microdepressions in the ground surface, where minerals transported in aeolian fashion and flowing down with precipitation water accumulate, and with them a suspension of a silty clay plastic fraction coming from post-coal waste. After settling at the bottom of the depressions, they seal them with an impermeable layer of clay sediments. The moist, clay substrate retains seeds transported with the wind and creates more favourable conditions for colonisation by pioneer plants in comparison with the neighbouring gravel and stone fraction habitats with completely unweathered material. Such results were found in changes in the fraction of soil particles and their effects on the stability of the soil-vegetation system in successions and restoration processes in degraded ecosystems [22,30,46]. The surface with initial succession has non-uniform morphology. Coarse, stony fractions are predominant, and the spatial distribution of the fractions often results from the non-selective way material is stored on the heap, which is characteristic of a majority of heaps [9,10,47].

Vegetation encroaching on these dead, often highly toxic, and dry substrates consists of usually pioneer species with low nutrient requirements, in addition to a strong and highly branched root system. By adapting in order to grow on a bare and exposed surface, they have an impact on changes in conditions and enable other, more demanding plant species to invade these areas [48]. The first pioneer species to colonise such a habitat are *Ch. palustre*, singly accompanied in a loose form by *O. biennis*, *E. canadensis*, *Tussilago farfara*, *Lepidium ruderale*, *Senecio viscosus*, *Linaria vulgaris*, and *D. carota* (Appendix A).

As a species initiating succession in the study area, *Ch. palustre* has a wide scope of ecological tolerance, i.e., a light-dwelling species, anemochorous diaspores with a highly developed flight apparatus, numerous flowers and associated high seed production, and a well-developed root system [49]. It develops best on rock rubble, scree, and gravel. Such conditions are also typical for coal heaps, and hence *Ch. palustre* is found en masse in habitats transformed by human activity. Analogous observations have been made in similar areas [50]. 

The initiation of primary succession has a completely different character in areas associated with the occurrence of an algal crust (Figure 1A (4)). This is a substrate consisting mainly of silty clay material (Table 1) deposited at the foot of the heap, as a result of surface runoff, where an organic mineral crust forms (as initial horizon) after stabilisation. The substrate material in this habitat is characterised by a larger sorption surface and a high water-holding capacity [51,52], and it creates habitats for successive species with different habitat requirements. After prolonged drought, this crust breaks down and morphologically resembles the features of vertisols (Figure 1A (4)).

The second stage of succession is mainly initiated by the tall grass *C. epigejos*, characterised by high ecological plasticity, a wide range of habitat and nutritional adaptations, and the production of a large number of light and volatile seeds that facilitate dispersal. 

Its extensive root system supports the absorption of nutrients not only from the ground but also from atmospheric precipitation [53]. The rapid colonisation process in coalmine spoil heaps leads to turf formation on the soil surface, which in turn inhibits erosive processes, also slowing down the recruitment of other plant species. *C. epigejos* is distinguished by a vegetative life-strategy type that contributes to its rapid growth in large areas [54,55]. In most heaps and post-industrial areas in Poland and Europe, this species initiates habitat-forming processes by creating thick turf while inhibiting biodiversity [16,18,50]. The annual biomass production of this species leads to an increase in the amount of organic matter on a poor substrate surface after plant decomposition. In subsequent years, it is able to independently create an organic level consisting of this species detritus, which is important in enriching the soil and, consequently, providing nutrients to other species [22]. The phase and stages involving *C. epigejos* can be described also as the sodding stage, because they stabilize the loose ground. The long, extensive (spread) root system of this species binds loose fine-grained material and thus facilitates soil stabilisation and further succession and soil-forming processes in the heap area [22,55]. It is a fast-growing and deep-rooting plant, as its roots reach a depth of up to 2 m, with long and strong rhizomes (up to several metres long) and with culm up to 200 cm high and with 2–4 joints [53]. Clonal diversity depends on habitat conditions [56], and it is especially significant in areas heavily loaded with trace metals. Easy build-up of rhizomes and rapid expansion were also observed in the anthropogenic habitats of coal-waste dumps in Upper Silesian region.

*C. epigejos* is a generalist plant with a wide ecological amplitude in terms of soil humus, nutrient content, soil fraction, soil moisture, and pH (1.9–8.5 in KCl). Such a range of pH was also observed in the area analysed (Table 2). Consequently, it is able to grow in a wide range of diverse habitats, both natural and those created by humans. *C. epigejos* can provide nutrients from soils by means of an extensive system of below-ground organs and the ability to use rhizomes for selective spreading of ramets into nutrient-rich patches [48,57]. In addition, an elevated atmospheric N deposition has been identified as a possible cause of the increased abundance of *C. epigejos* over recent decades [58].

The process and rate of spontaneous succession in the area analysed bears a similarity to other coal heaps in Europe [4,6,18,22,23,59] in comparison with intentionally reclaimed areas [16].

The floristic composition in this vegetation community was diverse in terms of habitat requirements resulting from the habitat mosaic. Hence, there are representatives of early successional species (*Hieracium pilosella Calamagrostis epigeios*, *Cerastium semidecandrum*, *Agrostis canina*) and late successional, typically forest species (*Deschampsia flexuosa*, *Melandrium album*). On unreclaimed coal heaps, the period of forest formation varies and may last several decades [16,60]. Moreover, this process may be limited by the toxicity of the waste, the thermal activity of the heap [9], the instability of the soil, and the strongly salinity or high acidity of the stored material [15]. Considering the period of heap formation, its management, and the areas subject to spontaneous succession, the flora composition reveals similarities to the heaps described by other authors in terms of floristic composition [15,16,19,21,22].

### 4.2. Soil Properties on the Coal-Waste Dump

The dump stores waste rock from underground workings, making the deposits accessible for exploitation, as well as waste generated from washing and cleaning the excavated material. The waste is stored non-selectively, and the stored material is unburnt. In terms of petrographic composition, mining waste is mainly clay and sand shale, with sandstone of different grain size and clay/carbonate binder also present in small amounts [28,51,61].

Mine-related anthropogenic soils are distinctive in that they are derived from unique parent materials in the form of mine spoils (tailings). Mine spoils are anthropogenic sediments comprised mainly of fragments of unweathered sedimentary rock that have been abraded, fractured, and rapidly exposed by blasting and excavation. These blasted rock fragments may be present in a wide variety of particle sizes, ranging from boulders 1–2 m in diameter, to fine silt and clay. Mine spoils are typically composed of 60% coarse fragments, and, if uncompacted, are characterized by unusually large voids not seen in natural soils [34,35].

A prominent feature of pedogenesis is the rapid weathering of coal refuse [34]. This material is characterised by high variability in physical and chemical properties and very low biological activity [60]. Post-coal waste in heaps has a very different granulation, and the weathering processes causes it to change within the surface layer and increase the proportion of finer fractions. It is also characterised by high salinity (especially on freshly dumped heaps), significant pH variability, carbon content, and low heavy metal content [51,62]. The waste is rich in assimilable magnesium and potassium compounds and poorer in calcium compounds [22].

The soils analysed did not differ fundamentally from similar soils developed on materials of carbon origin. They had similar reactions [28,63,64], nutrient content [22], and heavy metal content [65,66]. The principal differences between these soils stem from the different ages of the heaps and the type of reclamation work. On older heaps, the effects of pyrite transformation manifested in the form of a change in the soil reaction towards acidification are clearly visible [67]. In the soils investigated, pedogenetic processes are clearly observed in the form of organic (O), organic mineral (A) horizon forming, despite the short period of pedogenesis. The rate and processes of soil formation are also significantly influenced by the adjacent forest ecosystems, providing organic matter (in the form of plant litter) and propagules that accelerate the processes of vegetation colonisation [32]. The effect of vegetation development is observed in such ways as in the formation of turf and inhibition of water and aeolian erosion processes [25].

### 4.3. Soil Organic Matter Differentiation

Usually the predominance of odd-over-even numbers in the range *n*-C_25_–*n*-C_29_ is due to recent vegetation input, particularly epicuticular waxes of higher plants [68,69]. The presence of δ- and γ-Tocopherols may point to increased bioactivity and can be considered a good bioindicator in soil-forming processes. Tocopherols are synthesized only in photosynthetic organisms and act as protective components known for their antioxidant activity [70].

5β-Cholestan-3β-ol (Coprostanol) has been found in sewage discharge [71,72] and also in the filamentous green alga *Spirogyra longata* [73], and occurs in higher concentrations compared to other sterols.

Cholest-4-en-3-one is a cholesterol derivative and naturally occurring substance found in both plant and animal tissues. Its presence may result from biosynthesis or be due to the autoxidation of cholesterol [74]. Sterols are essential dietary nutrients for most aquatic invertebrates whose composition can differ markedly in microalgae. They are essential components of membranes to maintain the stability of cellular lipid bilayers [75]. The sterol composition of animals is usually dominated by Cholesterol. However, sterols in plants and algae are highly diverse [76,77,78,79].

A typical terrestrial higher-plant phytosterol biomarker is Stigmast-5-en-3β-ol (β-Sitosterol) [72]. Identifying these compounds can help to describe bioactivity during soil forming and also in algal crusts, e.g., the presence of 4-Stigmasten-3-one is the product of β-Sitosterol fermentation [80]. Stigmastanol is a plant stanol ester reduced from Sitosterol that resembles Cholesterol in its structure [81,82]. Ergosterol is present only in algal crusts, as it the main component of fungal membranes [83]. 

Styren is commonly present in the resins of trees and plants, and it is possible that styrene is a product of the biodegradation of naturally occurring compounds of Cinnamic acid, Cinnamic aldehyde, Cinnamyl acetate, Cinnamyl alcohol, Cinnamyl benzoate, and Cinnamyl cinnamate [84].

Methoxyphenol and Methyl-oleanonate are common in angiosperm lignin and the kingdom of plants, and like Friedelan-3-one, Benzoic acid (lignin degradation product) has been identified as a common marker of recent vegetation input [85]. The chlorophyll degradation end-product Methylethylmaleimide [86] was identified in all samples as a good marker of bioactivity in the samples, especially in bare coal wastes. The presence of isoprenoid thiophene, i.e., 3-Methyl-2-(3,7,11-trimethyldodecyl)thiophene, or alkyl thiophene (3-*n*-Hexadecylthiophene) indicated the introduction of sulphur to the specific lipid moieties or to the corresponding functionalized alkanes [87,88]. The presence of thiophenes in algal crust samples was explained by Fukushima et al. [89], with C_20_ isoprenoid thiophenes being produced in fresh water under conditions where either sufficient evolution or inflow of H_2_S and chlorophyll occur. In the reaction, Chlorophyll-α-derived phytol (the ester-linked side chain of Chlorophyll-α) and H_2_S took part via Phytadiene intermediates, where sulphur was incorporated into the conjugated s-cis diene moiety of Phytadienes, which can be formed by dehydration of Phytol. The Trithiolane identified in soils and algal crusts is probably related to components with fungi [90]. This compound was previously identified by Nádudvari et al. [91] in river sediments, coal wastes from erosion gullies, or in self-heated coal-waste dumps.

Vanillin was identified in resin components, leaves, roots, stems, fruit, seeds, prairie grassland soils [92] and on coal wastes where initial soil formed [42].

Phytene is a reduction and dehydration product of the Phytol side chain of chlorophyll-α [93]. Neophytadiene was identified as being one of the terpenes, and had previously been detected in several plants and microalgae [94,95]. Phytadienes are degradation products (acidic dehydration) of Phytol, the ester-linked side chain of Chlorophyll-α [89].

Such alkanols as 2-Dodecen-1-ol, 1-Hexadecanol, 1-Octadecanol, and 1-Icosanol were also identified by Abdel-Aal et al. [73]. The *n*-C_17_–*n*-C_24_ alkenes were reported in leaf cuticular waxes, insect species, and freshwater green algae lipids [96]. The possible source of alkenes in algal crusts is probably related to converting fatty acids into alkenes [97].

In many samples, a possible source of Benzaldehyde is the oxidation of Toluene [98]. Acetophenone and Indene were previously found in coal-waste samples or in self-heated coal waste by Nádudvari et al. [42] and Nádudvari et al. [91]. They are commonly present in fossil fuels, coal combustion, and coal tars [99,100].

### 4.4. The Contents of Heavy Metals in Soil and Plants

Similar results for post-mining waste, both in terms of the content of major elements (Ca, K, Mg, Na, Al, Fe, P, and S) and heavy metals (Z, Pb, Cu, Ni, Co, As, Cd, Hg, and Cr), were obtained in Poland by Spychalski [101], Klojzy-Kaczmarczyk et al. [102], and Klatka et al. [62]. Xin et al. [103], conducting research in Hunan Province, China, also reported similar values. The variation in elemental composition in heaps is influenced by granulometric composition, pH, and salinity [64]. The similarity in the chemical composition of post-coal materials is due to their genesis [104], and further changes are determined by geochemical transformations and the influence of biotic factors.

According to the *Geochemical Atlas of Upper Silesia* [105], the content of trace elements in the samples studied are similar to those in the surrounding soils. According to the Regulation of the Minister of the Environment of 5 September 2016 on the manner of conducting the assessment of the pollution of the earth surface [106], these results do not indicate that the norms of group “IV” constituting industrial areas, fossil lands and communication areas are exceeded.

Heavy metal content in the plant material was found to be within the proposed threshold values and did not exceed the limit values reported by other authors [107,108]. When generally assessing the level of heavy metal accumulation in the plants studied, it should be concluded that the degree is low, which indicates a lack of phytotoxicity of the post-coal materials in question. The findings of the analysis of the heavy metal content in the waste material tested here do not indicate an increased content of heavy metals when compared with the norms in force in Poland, industry guidelines, and literature. The content of heavy metals in the plant material collected from the landfill was within the proposed threshold values [104].

Compared to the other species analysed, *C. epigejos* proved to be of great importance for both soil enrichment and heavy metal accumulation in extreme environments. Bearing in mind *C. epigejos*’s habitat preferences and its ecological plasticity, the same plasticity was also expected on the physiological level. Several studies have investigated its behaviour in conditions of increased concentrations of heavy metals in the habitat [109,110], where significant changes in physiological characteristics were observed.

However, *C. epigejos* revealed certain potential in the field of phytostabilisation, considering the fact that it retains Zn, Cu, Cd, and Pb in its roots and reduces pollution through their binding. This is particularly important for the available pool of stated elements. The role of *C. epigejos* as a pioneer species inhabiting mining areas after environmental accidents was found to be important, since this species reduces the bioavailability and spread of contaminants into the environment [111] and also initiated soil-formation processes.

The chemical composition of *C. epigejos* contained such elements as Ca, K, and Mg (Table 3), and these are potential minerals that return to the soil after the plant dies at the end of the growing season. The significant phosphorus content (8600 mg-kg-1]) in the roots of this species indicates its organic origin [112]. In the roots of *Ch. palustre* and *C. epigejos*, the contents of some elements (Ca, Mg, P, Fe) are higher than in the above-ground parts of the plants, which is mainly due to the lifespan of the plant organ. Plant species that can grow in nutrient-poor habitats, like *C. epigejos* and *Ch. palustre*, usually have an ability to retain caught nutrients due to the longer lifespan of tissue [113]. The results obtained in terms of heavy metal content in the chemical composition of *C. epigejos* do not differ from those obtained by other authors from landfills of post-mining waste [62] and transformed by industry [114].

*R. pseudacacia* is the most common tree species in plantations in China and is effective in most greening of disturbed areas [115]. *R. pseudacacia* has been planted in gullies on China’s hilly Loess Plateau [3]. In Europe, this species is mainly planted along roadsides in both urban and rural areas, as it is resistant to environmental pollution [116] *R. pseudacacia* is widely used in vegetation restoration in many degraded areas [10,22], which is attributable to its rapid growth and nitrogen fixing in the atmosphere [117].

Nitrogen-fixing plants can affect the dynamic characteristics of the community, particularly for those habitats with poor nutrient surroundings [118,119]. On the other hand, the annual production of biomass (mainly plant litter: leaves, pods) by this species enriches poor or initial soil with mineral and organic matter, on which pedogenesis often depends. In the chemical composition of *R. pseudacacia* leaves, there are significant contents of Ca (12,400 mg∙kg^−1^), K (13,700 mg∙kg^−1^), Mg (3140 mg∙kg^−1^), and P (1350 mg∙kg^−1^). After decomposition, the elements return to the soil as potential nutrients, and become one of the most important sources of plant nutrients. In addition to nutrients, this plant also accumulates heavy metals [120]. The results obtained from *R. pseudacacia* branches in terms of their Cd content (0.18 mg∙kg^−1^) were several times lower than the values from post-industrial areas (2.48 mg·kg^−1^) reported by Palowski et al. [120]. Low contents of metals in the area of heaps in *Robinia* branches were found for Zn (14.8 mg∙kg^−1^), Pb (3.32 mg∙kg^−1^), and Cu (5.39 mg∙kg^−1^) in comparison with other results [119], where the content of Zn was on average 60.7 mg∙kg^−1^, Pb-29 mg∙kg^−1^, and Cu-9.69 mg∙kg^−1^. Similar regularities were established for the content of heavy metals in black locust leaves [116]. The bark of *R. pseudacacia* has also been found to be a good bioindicator of long-term cumulative traffic pollution, whereas leaves are good indicators of short-term seasonal accumulation trends.

The results obtained from post-coal waste in the Szczygłowice heap area with regard to heavy metal contents were in the following ranges: Zn 42–74, Pb 16.5–25.2, Ni 18.1–31.5, and Cu 26.7–40.2 mg∙kg^−1^. The findings of research conducted by Samecka-Cymerman et al. [116] on the urban environment of Oleśnica were characterized by ranges of heavy metal contents of Zn 132–181, Pb 30–99, Cd 0.6–1.9, Ni 8.9–17.8, and Cu 14–37 mg∙kg^−1^ in soil where *R. pseudacacia* develops.

## 5. Conclusions

The process of plant succession is closely related to the stages of coal-waste heap emplacement. The initiation of vegetation and soil development is determined by the variation in the grain size of the waste dump and the variation in the relief and the associated habitat mosaic. Algal crusts forming on silty clay, mineral organic material, accumulating in depressions in the area and at the foot of the heap, can be regarded as the focus of pedogenesis. The algal crust serves as a good environment for soil formation, as the first stage. The biomarkers separated the samples well, according to the bioactivity (soil presence) and compound degradation processes, where different organisms occupy the bare coal-waste material. Less favourable conditions prevailed in the adjacent habitats with a gravel and stone fraction and unweathered material.

Plant succession is initiated by species with a wide ecological tolerance to the area studied, such as *Chamaenerion palustre* (initial stage) and *Calamagrostis epigejos* (optimal stage). The important role of *C. epigejos* in initiating soil-forming processes and the importance of this species for soil enrichment and accumulation of heavy metals was also found. In a grouping dominated by *C. epigejos*, despite a short period of pedogenesis (c. 20 years), the formation of organic and humus horizons was observed.

The chemical composition of the soil material indicates a low degree of weathering of the disposed post-coal waste and homogeneity, while higher Ca and Mg contents may come from plant decomposition. The level of heavy metal accumulation in the plant and soil material examined was low. Their content in all analyzed soil profiles had similar values, indicating a similarity in the chemical composition of the disposed post-coal waste. Slightly higher contents of Zn, Pb, Ni, and Cu were recorded in the humus horizon. Their content is not reflected in the chemical composition of the plant species studied. 

As the results indicate, post-coal waste contains fewer heavy metals and does not exceed the thresholds defined by Polish law. Thus, these materials are characterised by a lack of phytotoxicity and the succession proceeds in a similar way to other anthropogenically transformed areas. These studies have the character of ecopedological documentation and may constitute comparative material for following the further course of vegetation development on the heap.

## Figures and Tables

**Figure 1 ijerph-19-09167-f001:**
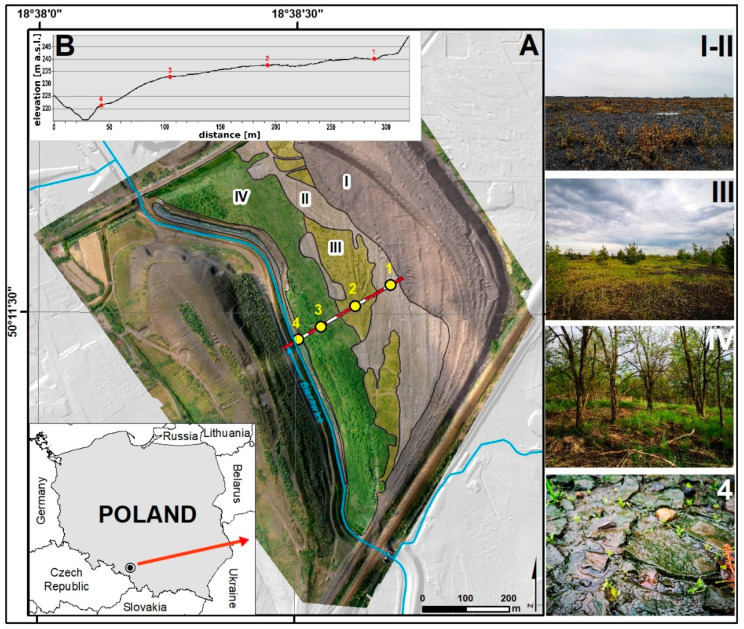
Location of the study area and sampling points: (**A**) (1–4)—soil sampling points and (I–II)—initial stages, III—transition stage, IV—terminal stages, 4—algae crust; (**B**) sampling sites in background of relief.

**Figure 2 ijerph-19-09167-f002:**
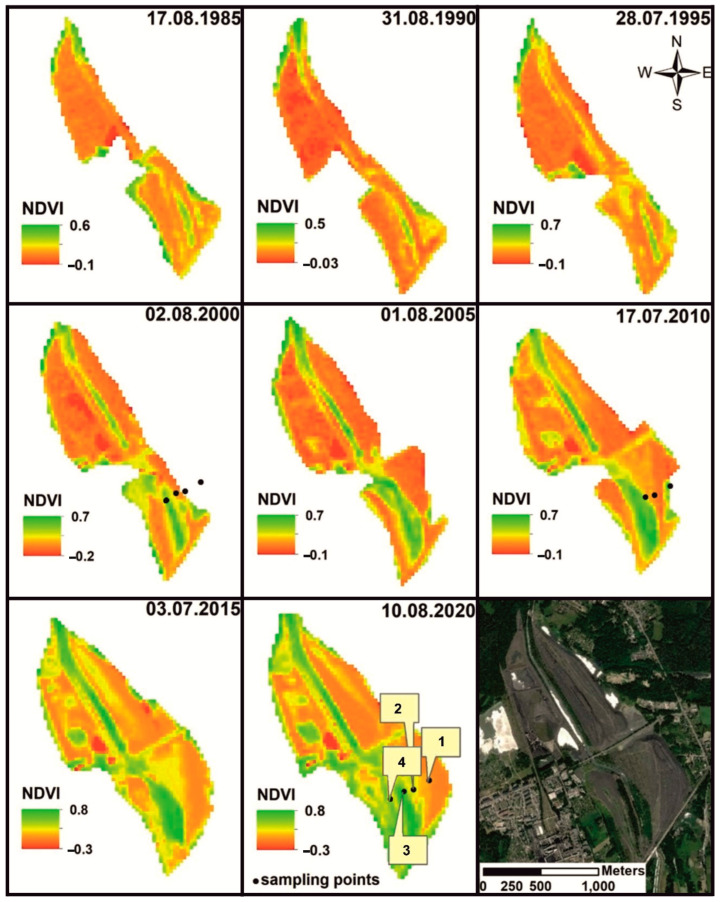
The dump developments and vegetation succession in during the last 35 years based on Landsat 4-5TM, Landsat 7ETM+, Landsat 8 OLI. Available online: https://earthexplorer.usgs.gov/ (accessed on 23 May 2021).

**Figure 3 ijerph-19-09167-f003:**
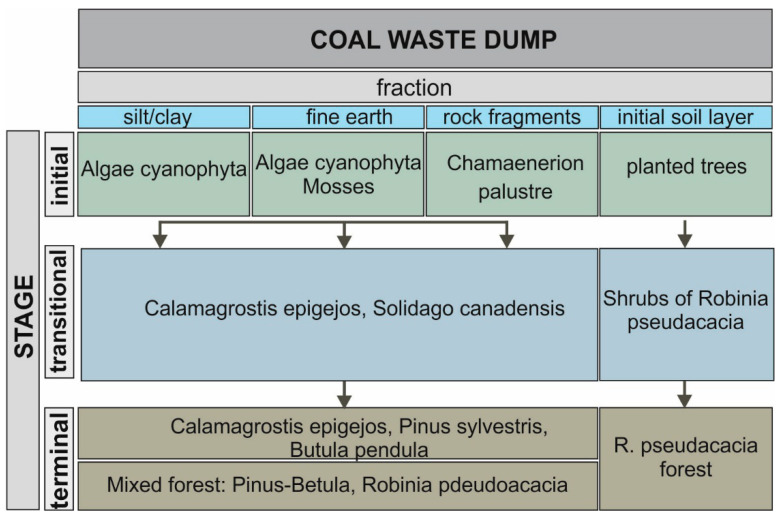
The course of the vegetation succession in the analyzed coal dump.

**Figure 4 ijerph-19-09167-f004:**
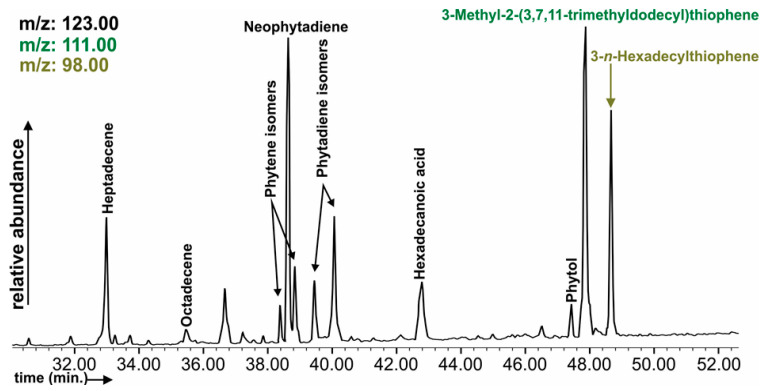
Representative chromatogram of organic sulphur compounds (OSC) and terpane-related compounds in profile 1 sample.

**Table 1 ijerph-19-09167-t001:** Percentage share of granulometric composition in coal dump.

		[mm]												
Horizon	Depth [cm]	>20.0	20.0–10.0	10.0–5.0	5.0–2.0	2.0–1.0	1.0–0.5	0.5–0.25	0.25–0.1	0.1–0.05	0.05–0.02	0.02–0.006	0.006–0.002	<0.002
		[%]												
Profile 1 Communities with *Chamaenerion palustre*														
C1	0–14	9.7	20.0	17.3	21.1	12.9	4.0	5.5	2.7	2.3	0.9	1.7	1.1	0.8
C1	14–90	4.1	22.4	20.6	21.9	13.8	4.6	5.1	2.0	1.4	0.7	1.0	0.9	1.5
Profile 2 Communities with *Calamagrostis epigejos*														
A	0–13	4.2	12.9	17.1	20.5	12.0	3.6	9.9	4.7	3.0	2.3	4.3	1.8	3.7
ACq	13–45	-	11.4	17.9	24.4	10.5	6.4	6.9	3.9	2.5	1.8	4.6	2.9	6.8
Cq	45–65	9.8	25.7	24.6	17.2	7.9	3.7	4.3	1.7	1.2	0.7	1.8	1.0	0.4
Profile 3 Communities with *Robinia pseudacacia*														
Olfh	4–0			3.7	12.1	4.2	4.5	8.6	10.2	8	16.8	13.6	7.2	11.1
A	0–23	-	16.3	11.7	14.1	16.0	2.6	5.7	4.8	3.0	5.5	5.5	4.3	10.5
ACq	23–60	-	13.7	20.9	27.5	14.4	5.6	5.5	2.5	1.6	1.9	2.3	1.5	2.6
Cq	60–80	7.5	19.1	20.9	22.4	5.7	2.2	3.9	2.7	1.7	3.2	3.4	2.2	5.1
Profile 4 Organic mineral crusts with green algae														
(1) OC	0.4	-	-	1.7	4.1	11.8	1.6	1.7	1.8	3.4	4.6	23.9	19.8	25.6
(2) OC	0.5	-	-	0.4	3.3	6.5	2.7	4.6	5.3	9.9	34.0	18.0	4.5	10.8
(3) OC	0.6	-	-	-	-	1.1	0.9	1.1	1.1	3.9	28.7	35.6	14.8	12.8
(4) OC	0.5	-	-	-	-	1.9	0.9	1.3	0.9	2.9	23.5	40.2	15.7	12.7
(5) OC	0.9	-	-	0.6	2.0	9.3	1.2	3.1	2.8	5.3	27.3	29.9	11.5	7.0

**Table 2 ijerph-19-09167-t002:** Physico-chemical properties of soils in the spoil heap during the succession.

Soil Horizons	Depth [cm]	pH		Loss on Ignition	Corg.	Nt	C/N	Mg Avail.	P Avail.	Pt	K Avail.	Hh [cmol(+)/kg]
		H_2_O	KCl	[%]				[mg·kg^−1^]				
Profile 1 *Chamaenerion palustre*												
C1	0–14	6.9	6.7	34.75	25.8	0.538	48	343.5	0.90	558.0	197.0	1.92
C1	14–90	6.6	6.5	34.14	31.2	0.316	99	498.0	1.00	572.0	249.0	1.84
Profile 2 Communities with *Calamagrostis epigejos*												
A	0–13	6.9	6.5	30.86	26.7	0.688	39	319.5	0.00	387.0	139.0	2.00
ACq	13–45	7.1	6.8	43.48	41.6	0.531	78	284.5	1.05	705.0	159.0	1.60
Cq	45–65	4.9	4.8	40.85	32.9	0.754	44	475.0	1.40	578.0	136.5	5.68
Profile 3 Communities with *Robinia pseudacacia*												
Olfh	4–0	5.8	5.3	46.5	26.4	0.751	35	299.0	0.70	440.0	191.5	4.36
A	0–23	4.8	4.1	18.73	10.7	0.884	12	326.0	1.40	196.0	143.0	5.36
ACq	23–60	5.5	5.2	32.84	27.5	0.402	68	357.5	1.92	281.6	132.0	3.76
Cq	60–80	5.4	5.1	24.37	14.5	0.706	21	393.5	0.60	318.0	143.0	3.92
Profile 4 Organic mineral crusts with green algae												
(1) OC	0.4	7.6	7.1	24.82	20.7	0.297	70	481.0	0.00	455.0	251.5	2.00
(2) OC	0.5	6.8	6.5	14.83	9.6	1.132	8	263.5	16.20	954.0	207.0	3.12
(3) OC	0.6	7.1	6.9	21.46	14.0	0.450	31	345.5	7.32	1588.0	199.0	2.32
(4) OC	0.5	7.1	6.8	22.47	16.1	0.446	36	507.0	7.40	1580.0	202.0	2.80
(5) OC	0.9	7.4	7.3	22.12	16.0	0.610	26	670.0	9.77	1310.0	277.0	2.04

**Table 3 ijerph-19-09167-t003:** Content of major elements in the plant materials and soil.

	Ca	K	Na	Mg	P	Fe	S	Al
	[mg·kg^−1^]							
plants parts	plant materials							
	*Chamaenerion palustre*							
aboveground	7000	6200	30	3480	710	310	2200	<100
roots	8400	5700	30	4040	530	710	1600	200
	*Calamagrostis epigejos*							
aboveground	1400	9700	20	850	1118	250	1300	<100
roots	1100	3200	110	760	8600	12,610	2200	4000
	*Robinia pseudacacia*							
leaves	12,400	13,700	3	3140	1350	170	2100	<100
bark	16,800	3300	2	510	320	210	1700	<100
roots	3800	11,600	5	1320	1030	2510	3000	1000
pods	2900	9300	<1	1470	2510	6	1900	<100
horizon and depth [cm]	soil materials							
	Profile 1 under *Chamaenerion palustre*							
C1 (0–14)	4200	1400	80	3100	580	15,900	3500	5400
C2 (14–90)	5700	1600	90	4500	530	18,100	5800	5500
	Profile 2 under *Calamagrostis epigejos*							
A (0–13)	6600	1500	90	4700	310	13,800	2600	6100
ACq (13–45)	3200	1600	90	2600	340	18,700	7700	5300
Cq (45–65)	1600	1700	90	1600	300	18,800	9600	5200
	Profile 3 under *Robinia pseudacacia*							
A (0–23)	1500	1600	70	2700	290	26,200	1300	6600
ACq (23–60)	1000	1600	70	1900	140	11,700	2100	5600
Cq (60–80)	1600	1700	70	2100	380	16,300	2300	6700

**Table 4 ijerph-19-09167-t004:** Heavy metals content in the plant material and soils.

	Zn	Pb	Cd	Co	Ni	Hg	Cu	As	Cr
	[mg·kg^−1^]								
plant parts	plant materials								
	*Chamaenerion palustre*								
aboveground	65.4	2.89	0.19	0.44	1.8	0.023	19.83	<0.1	2.3
roots	35.4	2.06	0.11	1.16	4.1	0.01	8.71	<0.1	2.1
	*Calamagrostis epigejos*								
aboveground	22.8	0.98	0.16	0.35	1.6	0.01	2.98	0.3	1.4
roots	70.0	21.16	1.12	17.50	25.9	0.193	37.53	10.3	9.8
	*Robinia pseudacacia*								
leaves	29.4	0.99	0.14	0.52	5.0	0.028	4.88	<0.1	1.7
bark	14.8	3.32	0.18	0.51	2.2	0.007	5.39	<0.1	1.7
roots	35.3	5.81	0.26	3.26	12.2	0.031	15.54	0.9	4.2
pods	27.7	0.21	0.08	0.77	10.1	>0.001	9.76	<0.1	1.8
horizon and depth [cm]	soil materials								
	Profile 1—with *Chamaenerion palustre*								
C1 (0–14)	49	23.6	<0.1	7.7	18.1	0.08	39.7	3.9	8
C2 (14–90)	42	21.2	<0.1	8.8	21.8	0.09	39.5	4.0	9
	Profile 2—with *Calamagrostis epigejos*								
A (0–13)	50	21.4	0.2	7.6	22.9	0.13	32.7	5.1	10
ACq (13–45)	44	21.5	<0.1	8.7	25.4	0.20	35.7	10.0	9
Cq (45–65)	46	25.2	<0.1	10.7	31.5	0.28	40.2	14.4	8
	Profile 3—with *Robinia pseudacacia*								
A (0–23)	74	17.2	0.2	8.5	25.5	0.09	26.7	4.8	14
ACq (23–60)	46	16.5	<0.1	8.7	24.3	0.07	31.7	2.4	10
Cq (60–80)	62	20.5	0.2	10.2	27.4	0.13	30.7	5.6	12

**Table 5 ijerph-19-09167-t005:** Content of selected major elements in the algal crust.

Sample	CRUST WITH ALGAE (Profile 4)							
	[mg·kg^−1^]							
	Ca	K	Na	Mg	P	Fe	S	Al
1 (OC)	5400	2100	570	4000	1100	19,100	2600	8700
2 (OC)	4800	1500	200	2800	1550	21,400	2700	8400
3 (OC)	9000	1800	220	4600	2460	28,800	3200	13,000
4 (OC)	12,700	1900	330	5500	2820	33,600	3500	14,400
5 (OC)	5900	2000	540	3800	1240	18,800	2400	9400

**Table 6 ijerph-19-09167-t006:** Content of heavy metals in the algal crusts.

Sample	CRUST WITH ALGAE (Profile 4)								
	[mg·kg^−1^]								
	Zn	Pb	Cd	Co	Ni	Hg	Cu	As	Cr
1 (OC)	277	41.8	1.4	15.9	31.9	0.12	52.9	6.9	23
2 (OC)	308	41.8	1.8	12.4	27.6	0.14	41.4	6.7	26
3 (OC)	649	66.4	3.4	16.9	36.3	0.18	63.7	11.0	44
4 (OC)	736	76.0	4.2	19.7	40.6	0.24	72.9	12.6	50
5 (OC)	319	38.2	1.7	13.0	29.6	0.11	42.6	6.8	25

## Data Availability

Not applicable.

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
