# Peer review of "Soil and Vegetation Development on Coal-Waste Dump in Southern Poland"

_ijerph, 2022, doi:10.3390/ijerph19159167_

Round 1

Reviewer 1 Report

The paper is very well written, and contributes the study in soil and vegetation development on coal‐waste dump , which describes the joint development of soil under the influence of vegetation succession against the background of environmental conditions.

There are some problems, which should be solved before it is considered for publication.

(1)In the article, the mechanism is mainly expressed qualitatively, and it is necessary to quantitatively reveal the part of the mechanism.

(2)I read the discussion section completely. The discussion section is the important part of a paper, but this manuscript mainly reported the cases in the literature through available publications. I recommend author adding several reasons for enhancing the reliability of deduction.

(3)Conclusions needs more in it, as it's more of an afterthought. In the conclusions, in addition to summarizing the results,please strengthen the explanation of their significance. The author need to highlight this paper's contributions. The authors are suggested to highlight important findings and include afterthought of this work.

(4)If possible, quoting some papers from our research group may also be considered

Author Response

Dear Reviewer,

Thank you very much for your constructive review on our manuscript (ID: ijerph-1806004) entitled “Soil and Vegetation Development on Coal-Waste Dump in Southern Poland". The paper has been fully revised according to the comments made by you.

We would like to resubmit the revised version for consideration of possible publication as a regular paper.

We would like to express our sincere appreciation to you for your constructive comments and the effort and the time You spent helping us to improve the quality of the paper. In the following, we provide a specific response to the comment, explaining how the paper is revised.

We have highlighted all the changes in our revised manuscript.

We hope that the revised manuscript addressed your concerns in a proper and satisfactory way.

Here are the responses to the Reviewer’s detailed comments and suggestions.

(1) In the article, the mechanism is mainly expressed qualitatively, and it is necessary to quantitatively reveal the part of the mechanism.

We considered that the research material and the results obtained were sufficient (also quantitatively reveal) to be presented in the form of an article. We do not deny the reviewer's suggestion that it is necessary to quantitatively reveal the part of the mechanism. Certainly, when preparing future articles on coal-waste dumping, we will express our results mainly quantitatively. This approach was determined by the interdisciplinary nature of the problem. In the case of quantitative analysis, each element could have constituted a separate article.  We would like to thank the reviewer for his valuable comments and will compulsorily take this suggestion into account in the future especially as we have quite a lot of material gathered in this matter.

(2) I read the discussion section completely. The discussion section is the important part of a paper, but this manuscript mainly reported the cases in the literature through available publications. I recommend author adding several reasons for enhancing the reliability of deduction.

We have related the discussion of the results to the available publications in relation to the hypotheses and objectives of the study.

(3) Conclusions needs more in it, as it's more of an afterthought. In the conclusions, in addition to summarizing the results,please strengthen the explanation of their significance. The author need to highlight this paper's contributions. The authors are suggested to highlight important findings and include afterthought of this work.

We agree with the reviewer that the conclusions lacked an emphasis on the significance of the results of the presented research. We have supplemented the conclusions with an explanation of the contribution of our article and they are included as a new paragraph in the Conclusion section.

(4) If possible, quoting some papers from our research group may also be considered

We would be happy to consider quoting articles from the Reviewer's research group; unfortunately we do not know the identity of the Reviewer. Therefore, we do not know what articles we might read.

Reviewer 2 Report

The paper is interesting and relevant, with dense and consistent information. Some points need to be clear, and comments were done along the paper. Then, in the form it is, it has no conditions to be accepted for publication but, after review the comments it may be reconsidered.

Author Response

Dear Reviewer,

Thank you very much for your constructive review on our manuscript (ID: ijerph-1806004) entitled “Soil and Vegetation Development on Coal-Waste Dump in Southern Poland". The paper has been fully revised according to the comments made by you.

We would like to resubmit the revised version for consideration of possible publication as a regular paper.

We would like to express our sincere appreciation to you for your constructive comments and the effort and the time spent helping us to improve the quality of the paper. In the following, we provide a specific response to the comment, explaining how the paper is revised.

We have highlighted all the changes in our revised manuscript.

We hope that the revised manuscript addressed your concerns in a proper and satisfactory way.

Here are the responses to the Reviewer’s detailed comments and suggestions.

Line 16:

The sentence has been rewritten as suggested and now sound as:

The mean range of pH (H2O) values in the profiles was:  6.75 ±0.21 (profile 1), 7.2 ±0.31 (profile 2), 6.3 ±1.22 (profile 3) and profile 4 is 5.38 ±0.42.

Lines 18 - 19:

The sentence has been rewritten to the clearer version and now sounds as:

The highest content of total nitrogen (Nt) was found (1.132%) in the algal crust as well as in sub-horizon of the organic horizon (Olfh-0.751 %) and humus (A-0,884) horizon in a profile 3 under the initial forest

Line 20:

The units has been corrected

Line 70:

The word „are” has been replaced with „were” as suggested

Lines 100 – 101

The year of sampling has been added

Lines 105 - 106:

It ran from the newest areas of heaps with freshly dumped post‐mining waste, through sodding (soddy horizon in soil science indicate the surfaces stabilized by grasses) areas.

 Line 122:

 The year of sampling has been added as suggested

Line 125:

The profiles were in the soil sampling point? Where the profiles were produced?

Yes, each point (in Arabic number) in Figure 1A represents a soil profile. The points (1-4) are soil profiles.

Line 128

Disturbed or undisturbed soil samples?  What depth were collect?

Soil samples were taken at the point where soil-forming processes occur. In general, the area has a disturbed character. For profiles 1, 2 and 3 the sampling depth depended on the depth of the root zone. In case of the algae crust, it was about a few mm, as the essence was only covered by a green coating. The depth of the profiles is given in Table 1.

Line 130:

How many sample for profile:

The number of samples varied from profile to profile, which was due to the stage of the soil profiles. This is due to the fact that samples were taken for the root zones, and this in turn is related to the succession stages. The number of samples taken refers to the soil horizons identified (see Table 1).

Line 176:

Numbers in Figure 2 has been corrected and numer 4 has been added

Line 244:

What is an Olf meaning?

The uppercase letter O denotes the organic horizon (master horizon) while the lower case letter indicates the degree of decomposition of organic matter within the main horizon. Ol- sub-organic horizon with undecomposed leaves, cones, barks and other; Of- poorly decomposed and plant tissues to be distinguished. If we write Olfh this means that the decomposition of the organic matter is non-uniform and the different fractions are in different stages of decomposition.

Olf= O- organic master horizon l-litter subhorizon, f-fermentation subhorizon

Lines 244 – 248

Units has been added

Lines 249 – 257:

Entire paragraph has been corrected exactly as suggested

Line 261:

Sentence corrected

Line 263:

Maybe a photography of each profile would be interesting

Unfortunatelly we are not able to add technicaly acceptable photographs

The granulometric analyse should be described in methodology

Information about methodology of granulometric analysis has been added to the Methodology

Line 275:

Gebhardt???

This is a method for the determination of total phosphorus of anthropogenic origin. We have added this information in the methods section.

Lines 270 – 278

All units has been modified

Line 279:

„Salt profile” corrected to „soil profile”

Line 288

„Loss Ignition” corrected to „Loss on Ignition”

Line 289:

What is the meaning of numbers?

The numbers indicate the number of algae crust samples. The numbers has been unified in Tables 1, 2, 3 and 6

Depth?

Yes, this is the depth of sampling in cm

Line 343:

Units has been corrected

Line 347:

Units has been added

Lines 354 – 358

Units has been corrected

Lines 364 – 365

„-” has been replaced with „=”

Lines 371 – 378

Units has been corrected till the end of the text

Line 399

Lead???

Corrected to Pb

Lines 824 and 823

Doubled numbers have been deleted